# New Digital Workflow for the Use of a Modified Stimulating Palatal Plate in Infants with Down Syndrome

**DOI:** 10.3390/dj14010026

**Published:** 2026-01-04

**Authors:** Maria Joana Castro, Cátia Severino, Jovana Pejovic, Marina Vigário, Miguel Palha, David Casimiro de Andrade, Sónia Frota

**Affiliations:** 1Faculty of Dentistry, University of Porto, 4200-393 Porto, Portugal; maria_joanacastro@hotmail.com (M.J.C.);; 2Center of Linguistics, School of Arts and Humanities, University of Lisbon, 1600-214 Lisbon, Portugal; catiaseverino@edu.ulisboa.pt (C.S.); jpejovic@edu.ulisboa.pt (J.P.); mvigario@edu.ulisboa.pt (M.V.); 3Child Development Centre DIFERENÇAS, 1950-392 Lisbon, Portugal; miguelpalha@diferencas.net

**Keywords:** stimulating palatal plate, digital workflow, oral-motor dysmorphologies, infants, Down Syndrome, speech auditory-motor link

## Abstract

**Background/Objectives**: Down Syndrome (DS) is frequently associated with oral-motor dysmorphologies, like oral hypotonia, tongue protrusion, short palate, and malocclusion, compromising the oral functions of sucking, chewing, swallowing, and speech production. Therapeutic interventions with stimulating palatal plates (SPP) have been proposed to prevent and improve oral-motor dysmorphologies in DS. This study proposes a new digital workflow for the manufacturing and use of a modified SPP. **Methods**: We report the application of the new workflow to five clinical cases, all infants with DS showing oral-motor disorders, aged between 5 and 11 months. The workflow is described step-by-step, from the mouth scanning protocol and model printing to SPP manufacturing and delivering, and assessment of oral-morphological features and language abilities via video captures and parental questionnaires. Key novel features include an SPP with an acrylic extension with a pacifier terminal and, importantly, the use of an infant-friendly intraoral scanner. **Results**: The new workflow had good acceptability by infants and parents, offering a safe, easy-to-implement, and feasible solution for SPP design, as it avoided the high risks associated with impression materials. It also supported the use of the SPP to promote tongue stimulation, retraction, and overall oral-muscle function in oral-motor disorders in children with DS, especially in infants. **Conclusions**: Within the limitations of the current study, it was shown that the proposed digital workflow constitutes a viable and infant-friendly approach to the production and use of a modified SPP, and thus promises to contribute to improving oral morphology and auditory-motor language abilities.

## 1. Introduction

Trisomy 21, or Down Syndrome (DS), is the most common genetic disorder among newborns. It is an autosomal aneuploidy, usually caused by a partial or complete duplication of chromosome 21 [1,2,3]. Worldwide, it is estimated that one case in 1000 births is a child with DS, and there were almost no changes in the prevalence of DS between 1990 and 2019 [4]. The prevalence of DS per 10,000 live births in the early 2000 was 15.74 in the United States [5] and 79 in Europe [6].

There are considerable phenotypic differences between individual patients with DS. However, there are specific features common to all. Regarding craniofacial characteristics, it is frequent to observe brachycephaly, relative macroglossia, craniofacial dysmorphia, oblique palpebral fissures and epicanthus, ear pinna malformations, and a flat nasal base [1]. It is well known that the underdevelopment of the midface leads to palatal atresia and a narrowed, elevated, and V-shaped palatal vault [1,7,8]. These, along with a protruding tongue, contribute to the establishment of malocclusions, such as anterior open bite, posterior crossbite, and a higher incidence of class III dental malocclusion [1,9].

The presence of hypotonic muscles in DS reveals an incomplete mouth closure, a protruding tongue, and increased oral breathing [1]. This imbalance causes oral-motor problems, compromising the oral functions of sucking, chewing, and swallowing [1,9,10,11]. It also steers the orofacial growth to develop malocclusion and craniofacial malformations, like midface hypoplasia. These oral-motor features also compromise speech [12,13]. Indeed, young children with DS typically have a high- and narrow-arched palate, and a relatively small jaw and mouth when compared to their tongue, which, together with hypotonic muscles, impacts sound articulation [11,14]. Moreover, recent research suggests that there is an auditory-motor link that influences speech perception, which emerges early in infancy [15]. For example, experimentally induced oral-motor impairments have been found to disrupt infants’ ability to discriminate speech sound contrasts. It is thus an open question to be empirically addressed whether the oral-motor problems in DS not only affect speech production, but also speech perception.

To prevent the development of malocclusion in children with DS, Castillo-Morales [16] developed a method of orofacial regulation therapy with the help of speech therapists and physiotherapists, which promotes muscle stimulation exercises, combined with the use of an intraoral appliance, named the stimulating palatal plate (SPP). This approach was used in several studies demonstrating improvement in tongue posture, lip closure, and oral-facial function in children with DS after the use of the SPP [8,17,18,19,20]. The SPP designed by Castillo-Morales was a removable orthodontic device to be used, preferably, before the eruption of primary teeth. It should be placed against the upper arch. It had a stimulatory area, the lingual stimulator, located on the lingual surface of the SPP. This lingual stimulator consisted of an acrylic button with a circular or oval form, approximately 8 mm in diameter. The oval form is used in children with tongue diastasis, while the round-shaped button should be used in children without it. A second stimulation area can be added—a vestibular stimulator, to encourage movements, if needed. It is made of small spheres or extra stimulation zones and/or tiny grooves or elevations located in the anterior and sublabial edge of the plate [17].

Building on Castillo-Morales’ work, Andrade proposed a modified SPP [21,22]. The modification consisted of adding an extension of acrylic to the SPP and a “pacifier-like” mouth shield. This modification addressed several issues with the traditional SPP and allowed for the extended use of the palatal plate. In particular, it promoted the acceptance of the SPP by the child, as well as parents’ confidence in its use, given the additional safety features offered by the pacifier mouth shield. The additional safety allowed for the use of the SPP also during the night, without the risk of losing the plate, swallowing, or choking. This also promoted patient compliance with the device. A comparison between the effects of the traditional SPP and the modified SPP showed that the modified SPP not only provided similar results to the traditional SPP but also offered better outcomes. Although no significant differences were found in most parameters, two parameters displayed significantly better results with the modified SPP: mouth semi-open or open with the tongue on top of the lower lip or pressing it, and open mouth with tongue inside the mouth [21]. Thus, in the present study, the modified SPP proposed by Andrade [21] was used and will be referred to as pacifier SPP (PSPP).

To manufacture the SPP, it is necessary to take impressions of the upper arch. Hence, the impression procedure is a crucial step in the production of the device. The method that was conventionally implemented used a quick-setting material to obtain a plaster model. The SPP was then designed over this plaster model. However, the conventional method has possible associated complications. According to Chate [23] and Behera et al. [24], this type of procedure may cause the aspiration of fragments of the impression material, airway obstruction that might be partial or complete, cyanosis, and the development of erosion or infection. In order to reduce the overall risk for children, and infants in particular, the workflow must be adapted. The introduction of intraoral scanners as a feasible and safer procedure for neonates and infants with craniofacial malformations has been documented in the literature [25,26]. Given the benefits observed, in this study, we explored using intraoral scanners for producing the modified SPP.

This study proposes a new digital workflow for the construction and use of a modified SPP. Our proposal combines the modified pacifier SPP with a digital workflow based on the use of an intraoral scanner device, thus addressing important drawbacks that characterized conventional approaches. Furthermore, it introduces a way to assess developing language abilities during the use of the palatal plate, resorting to parental questionnaires, as an additional novel feature of the proposed workflow. Our central goal is to evaluate the feasibility and acceptability of the proposed workflow, aimed at facilitating the production and use of this therapeutic device, with potential implications for the promotion of oral morphology and functionality, as well as auditory-motor language abilities in infants with oral-motor dysmorphologies.

## 2. Materials and Methods

The study was conducted as part of the P2LINK project (Perception-Production Link in Early Infancy: A language acquisition oral-motor intervention study), at the Lisbon Baby Lab of the Centre of Linguistics of the University of Lisbon. The research was conducted in accordance with the recommendations of the European Union Agency for Fundamental Rights and the Declaration of Helsinki, and was approved by the Ethics Committee of the School of Arts and Humanities of the University of Lisbon (19_CEI2021). Informed written consent, including informed consent for publication, was obtained from all the participants’ legal guardians.

### 2.1. The Clinical Cases

Five children with DS were included in the study. The five children were referred to the P2LINK project by the clinical staff of the Child Development Center—Diferenças, and constituted the first group of children that participated in the P2LINK project, representing varied ages and sexes. Participants 1 and 2 were 5-month-old girls, Participant 3 was a 7-month-old boy, Participant 4 was an 11-month-old boy, and Participant 5 was a 6-month-old girl. They were all born full-term. Only Participant 4 had other associated comorbidities (with deglutition problems and a collapsed lung after cardiac surgery, leading to the use of a nasogastric tube and oxygen tube until 10 months of age). On examination by a speech therapist and a pediatric dentist, all five infants presented tongue protrusion, hypotonia of the upper lip, and an open mouth with protruding lower lips (Figure 1a–e). These oral characteristics are indicators for the use of the SPP [8,16,17,18,20,27]. Additionally, three of the infants had never participated in any language intervention or speech therapy prior to or during the study, whereas the other two (Participants 3 and 4) had oral-motor stimulation provided by speech therapists during the study. This factor would be taken into account in future analyses of infants’ developing language abilities. The inclusion of a small and diverse sample of participants was driven by the need to examine infants with varied characteristics while keeping the study viable to be concluded in time to adjust the workflow, if necessary, before other infants were involved. Thus, a small and diverse sample of participants provided a good first test of the feasibility and acceptability of the new digital workflow.

### 2.2. Materials

A modified SPP was used. The pacifier SPP (PSPP) exhibited a lingual-stimulating button in the lingual surface of the plate, and included an acrylic extension that terminates in a mouth shield with a handle (Figure 2). The stimulating button was designed to regulate tongue position by repositioning the tongue within the oral cavity and to increase tongue activity. The extension with the mouth shield, besides preventing the infant from swallowing or choking on the plate, promoted the closing of the mouth behind the shield and facilitated nasal breathing. The angle and length of the extension were adjusted according to the lip and mouth occlusion features of each child. Overall, the PSPP was designed to improve oral-motor functions.

For the impressions of the upper and lower arch, a digital intraoral Dexis™ IS 3800 W scanner (Dental Imaging Technologies Corporation, Hatfield, PA, USA) was used. This is a small, light, and wireless intraoral scanner (IOS) device.

The use of the PSPP was monitored through video registration sessions and language and communication assessments. Video registration sessions, a common procedure in interventions with a palatal plate, took advantage of personal and home digital devices owned by the infant’s family, such as smartphones, iPads, and computers, and were implemented through a communications platform that allows for audio–visual connection between users. The video recordings had two main goals: (i) to monitor the use of the PSPP, measure the infant’s and parents’ acceptability of the plate, and identify the need for additional clarification or adjustments; and (ii) provide data for the assessment of oral-morphological variables. In addition, two parental questionnaires were used for the language and communication assessments: The Communication and Symbolic Behavior Scales Developmental Profile (CSBS-DPTM) Infant-Toddler Checklist, a screening tool that measures the communicative, social-affective, and symbolic development of infants between 6 and 24 months of age [28]; and the MacArthur–Bates Communicative Development Inventory (CDI) Short Forms, which assesses infants and toddlers’ early language abilities [29]. Parents’ response to the questionnaires was obtained via online forms. The questionnaires had two goals: (i) to evaluate the infants’ developing language and communication abilities before, during, and after use of the PSPP; (ii) to measure the parents’ acceptability of the overall workflow through their engagement in responding to the questionnaires.

### 2.3. Procedure

A flowchart showing the new digital workflow is provided in Figure 3. The step-by-step application of the workflow is described below, from the mouth scanning protocol and model printing to PSPP manufacturing and delivery, and lastly, the assessment of oral-morphological features, language abilities, and acceptability of the PSPP.

#### 2.3.1. Mouth Scanning and Model Printing

Mouth scanning of the infants was conducted using the digital IOS Dexis™ IS 3800 W (Dental Imaging Technologies Corporation, Hatfield, PA, USA) to register both jaws. Digital impressions were registered by the DEXIS™ IS ScanFlow (version 1.0.11.11, Dental Imaging Technologies Corporation, Hatfield, PA, USA). During scanning, the infants were comfortably lying down on a baby recliner chair, and a pediatric dentist positioned herself in front of them. A research assistant helped with the process.

A standard scanning protocol was used in order to maximize scan efficiency. Scanning started at the upper jaw, in the left maxillary tuberosity, and the head of the scanner was moved along the alveolar arch, passing the incisive papilla up to the contralateral maxillary tuberosity. From here, the palate was scanned, and finally, the vestibule area of the maxilla, including the labial and buccal frenula. In this area, the scan was tilted in its vertical axis. After that, the lower jaw was scanned, starting at the left retromolar area. The head of the scanner was placed in the lingual side and ran along the mandibular alveolar arch until the right retromolar area was reached. Then, the scanner was rotated, and the buccal side was scanned, following the opposite path. If the scanner lost its scanning position, the incisive papilla or the last successfully scanned areas were retaken as a starting point. Digital impressions for three of the infants are shown in Figure 4.

The removable head of the intraoral scanner was changed between patients and cleaned using a thermal disinfector (Autoclave class B). After that, it was packaged in a sterilization sleeve.

After acquiring the raw scan, the DEXIS™ IS ScanFlow post-processing tool calculated the surface of the scan. The scan file was saved in the Polygon File Format (.PLY) and Standard Tessellation Language format (.STL) and sent to the laboratory. Then, the files were exported to the EXOCAD/Model Creator design program, removing and smoothing out possible gaps in information created by the intraoral scanner, thus obtaining a new STL file. This new STL file was printed with the Microlay Microform software (Version 1.1.9.6, Microlay Dental 3D Printers, Madrid, Spain), which allowed us to define the best strategy for printing each model individually. The printer used was Microlay Versus 385 (Microlay Dental 3D Printers, Madrid, Spain), a precise 3D printer with a 65-micron horizontal resolution (X, Y) and a 10-micron vertical resolution (Z), capable of high accuracy and repeatability. The acrylic used to print the model was Microlay’s Printfit Dental Model (Microlay Dental 3D Printers, Madrid, Spain), with a thickness of 385–405 nm. The digital models of the maxilla and mandible for Participant 1 are shown in Figure 5.

#### 2.3.2. Manufacturing and Delivery of the PSPP

After printing the models, the cleaning and polymerization process continued. Cleaning was performed by placing the models inside an ultrasound device for 5 min, with 99% isopropyl alcohol. Once the cleaning procedure was complete, polymerization followed in the Otoflash G-171 n2 Photopolymerizer (Microlay Dental 3D Printers, Madrid, Spain), with nitrogen on the upper face of the model, and then repeating the process for the lower face of the model. The function of nitrogen was to prevent the printed model from coming into contact with oxygen during photopolymerization, thus eliminating the dispersion layer from the model’s surface.

The PSPP consists of two parts: the body and the handle. The base of the body was created from the physical model using Dental Flux 060 Clear 1.5 mm (Dentaflux, Madrid, Spain), with the help of a thermo-vacuum machine. The created base was then cut and smoothed manually. For the construction of the stimulating button, a self-polymerizing acrylic Weropress^®^ (Merz Dental GmbH, Ladenburg, Germany) was applied to the PSPP’s base, with mechanical retention, and polymerized at 45 degrees Celsius for 25 min with 2.35 bars of pressure (according to the manufacturer’s instructions). This procedure allowed us to obtain the union between the self-polymerizing acrylic and the thermo-vacuum base plate. Both parts were then smoothed and shaped (Figure 6). The materials used for the base are marked by manufacturers as biocompatible. The official data sheet of Dental Flux 060 Clear 1.5 mm (Dentaflux, Madrid, Spain) states the chemical identity as a polyester (terephthalic acid polyester—i.e., PET family) and features CE declarations, ISO 13485 status (company level), and a CE Declaration of Conformity for products—all of which cover regulatory conformity. In turn, the Weropress^®^ (Merz Dental GmbH, Ladenburg, Germany) is described as a cold-curing, methylmethacrylate-based (PMMA) resin system, used for denture bases, partial/full dentures, and splints, conforming to EN ISO 20795-1 (Type 2, Class 1). For the handle, the teat and rim of a conventional pacifier were cut, leaving the ring. A 1.75 mm × 0.90 mm orthodontic wire (Remanium^®^, Dentaurum GmbH Co., Ispringen, Germany) was bent and mechanically inserted in the ring and onto the body. The wire was then coated with a self-polymerizing polymethyl methacrylate (PMMA). For the final finishing of the PSPP, polishing with pumice stone and high-gloss paste specifically for acrylic was used.

After a 10 min video session of the infant facing the camera and in profile position (with a duration of 5 min for each position), the PSPP was delivered to each patient. It was placed in the mouth of the infant, applying pressure on the stimulation button, with the index finger, for 20 to 30 s to ensure a vacuum. After the vacuum was established, a pediatric dentist performed the following checks: (1) verified whether the base of the PSPP was well-seated on the mucosa, without areas of looseness or excessive compression; (2) applied alternating pressure to different areas of the PSPP, checking for tilting or displacement; (3) checked for peripheral sealing, verifying if the edges of the PSPP provided adequate sealing preventing air from entering and ensuring retention; (4) observed whether any area of the acrylic caused injury to the mucous membrane, and if so, wore it down; (5) inspected whether the lips were sealed, and if not, the PSPP tip was readjusted until lip seal occurred; (6) checked whether the tongue was stimulated to remain inside the oral cavity; (7) observed whether a gag reflex occurred, and if so, the acrylic was worn in the most posterior area of the palate and the height of the stimulation button was reduced; (8) verified whether occlusion occurred, ensuring the mandible made contact with the whole body of the PSPP, and if not, wearing it down until it did. This set of qualitative measures of PSPP fit was performed following standard procedures with stimulating palatal plates (e.g., [21]), given that, to the best of our knowledge, there are no non-invasive quantitative measurements specifically tested for this clinical population at such a young age. When adequate retention of the PSPP was confirmed, photographs and videos were taken with frontal and lateral views.

The recommendation provided was to use the PSPP, at first, for short periods (i.e., five to ten minutes, two to four times a day). As the child adapts to the device, we proposed an increase in the time of use to 15 to 60 min, four times a day. To achieve the most benefit from the PSPP, the suggestion was to use it for as long as possible, including when the child sleeps, not exceeding 15 h of use per day. Figure 7 illustrates the use of the PSPP.

#### 2.3.3. Assessment of Oral-Morphological Features, Language Abilities, and Acceptability of the PSPP

During the time of infant use of the PSPP, video recordings were made of all the clinical cases in separate sessions, following commonly accepted procedures in palatal plate interventions [18,20,21,30]. After the initial baseline session before the use of the PSPP, the infant and parents were engaged in participating in monthly recording sessions that included 5 min video samples of the infant without and with the PSPP, facing the camera and in profile position, with the mouth area clearly visible. The recordings allowed for the assessment of oral-morphological features, such as tongue position variables (intraoral, intermediate, extraoral) and mouth posture variables (closed, semi-open, open). Importantly, the recordings provided crucial information on the time and frequency of use of the PSPP, and on infants’ and parents’ overall acceptability of the plate. When it was not possible to conduct a video recording, a contact was made by phone to collect the information.

After an initial baseline assessment before the use of the PSPP, the developing language and communication abilities were assessed individually every three months, through the CSBS-DP Infant-Toddler Checklist and the CDI Short forms. The regular assessment of language abilities is an innovative feature of the present workflow, enabling a future exploratory study of the potential impact of oral-motor problems in DS on the auditory-motor link behind speech perception and production. Moreover, given that both infants who received concurrent language intervention/speech therapy and infants who only used the PSPP without any speech therapy participated in the study, future analyses should determine any group differences and whether PSPP use might lead to potential language improvements per se and/or result in added value. Parents were contacted to fill in electronic forms for the CDI and CSBS-DP, using secure web-based tools. The parents’ responsiveness to the language questionnaires was also taken as an indicator of the overall acceptability of the PSPP workflow.

#### 2.3.4. Data Analysis

Given that the aim of the present study is to put forward a new digital workflow for the use of a modified SPP, we focused only on the data variables that provided information on how safe, easy to implement, and feasible the new workflow was, as well as on the infants’ and parents’ acceptability of the PSPP workflow. Specifically, on the basis of data from the five clinical cases, we conducted a descriptive analysis of (i) the viability and acceptability of the mouth scanning, (ii) the speed of manufacturing the PSPP, and (iii) the acceptability of the PSPP by infants and parents. For (iii), we considered frequency and overall time of use of the PSPP, comments from parents, and their response rate to the follow-up assessment sessions and questionnaires.

## 3. Results

The data for the relevant variables under analysis are presented in Table 1. Additional (raw) data is provided in the Appendix A (Table A1).

### 3.1. Mouth Scanning: Viability and Acceptability

Each scan was completed in a single session, taking an average of 13.4 min to complete (Table 1). The median scanning duration of the upper jaw was 8.2 min and 5.2 min for the lower jaw. A major advantage of using intraoral scanner technology was the receptivity of parents, given the low risks and infant-friendliness of the small IOS device. Nevertheless, the operator noted the following challenges: (i) the dimensions of the head of the scanner, although small, sometimes made it difficult to move in the small oral cavities of the infants under study; (ii) children with DS present muscular hypotonia and a protruded tongue, which may add to the difficulty of maneuvering the head of the scanner; (iii) the increased salivary production characteristic of children with DS caused some interruptions in the scanning process, due to a lack of reading by the scanner; (iv) the sucking or biting reflex combined with the rigidity of the scanner’s head occasionally led to minor and superficial injuries to the gingiva and pauses in the scanner’s reading.

The relatively short duration of the scanning facilitated both the infant’s and parents’ collaboration. However, in the case of Participant 2, due to a longer scanning time, the infant became irritable and cried, forcing the procedure to be interrupted so that the infant could be calmed down.

Despite the very young age of infants, the specific traits of this clinical population, and across participant variability, the use of the digital IOS device for mouth scanning was viable in all cases and reached good acceptability from infants and parents.

### 3.2. From Mouth Scanning to Delivery of the PSPP

The time between the scanning and the delivery of the PSPP was 19 days on average (Table 1). This interval included all the exchanges between the Lisbon Baby Lab and the manufacturing lab, as well as the scheduling of the delivery according to the parents’ and the pediatric dentist’s availability. The PSPP fitted well in all infants. The final fit was assessed through the set of qualitative measures described in Section 2.3.2. In all cases, adequate retention was achieved, with the absence of tilting or displacement movements, good peripheral seal, occlusion, lip seal, and adequate tongue position. When placing the PSPP, at first, there was an automatic closure of the mouth and placement of the tongue inside the oral cavity in all children (Table A1). Then, there was an increase in saliva production and consequent drooling. In the case of Participant 5, this increase in salivation caused some gagging. After a few minutes of using the PSPP, Participant 1 and Participant 2 began to induce vomiting in an attempt to remove the PSPP. This reaction was later stopped after calling the child’s attention to not do so. Chairside adaptations of the PSPP were not necessary. Importantly, after 10 min of use, all infants showed good adaptation and acceptance of the PSPP, with the PSPP remaining lodged in the upper maxilla for the full 10 min.

### 3.3. Acceptability of the PSPP by Infants and Parents

Frequency and overall time of use of the PSPP are reported in Table 1, by participant. The overall time of use varied between 6 and 11 months, and the frequency of use per day varied from between 5 and 15 min to between 1 and 6 h.

Participants 2, 3, and 5 discontinued PSPP use due to the eruption of primary teeth and consequent misfit of the PSPP, allowing for the infant to remove it. For Participants 1 and 4, the first PSPP became unfitted due to the transverse growth of the upper jaw and consequent ease in removing the “pacifier”, as well as primary teeth eruption. A second dental scanning was conducted, and a second PSPP was manufactured, as both the child and the parents were willing to continue to use the PSPP. Additionally, for Participant 4, in a subsequent appointment, it was necessary to wear the acrylic with a drill in the eruption area of the first upper molars. This allowed for a complementary use of the PSPP for two more months.

The workflow of PSPP use included periodic contacts and responses to questionnaires. Parents responded actively to the monthly contacts and to the trimonthly language questionnaires, as shown in Table 1. The rate of parents’ response to follow-up contacts was calculated considering the number of contacts divided by the total duration of PSPP use in months. The mean response rate to all contacts (by phone and video) was 94%, with all participants above 75% and four above 90%. However, it was not always possible to carry out the video sessions, and the mean response rate to the video contacts was lower (67%), with large variation across participants. Participant 3 only completed half of the video sessions, whereas Participant 1 completed more than 4/5 of the sessions.

To compute the response rate to questionnaires, a denominator was obtained by the division of the total duration of PSPP use in months by three, given that the questionnaires were sent every 3 months. The rate of parents’ response to the language questionnaires was then calculated, considering the number of questionnaires completed divided by this denominator. Both the CSBS-DP and the CDI had a 100% response rate (Table 1). For participants 2, 3, and 4, parents provided more responses than those strictly required (Table A1), as they sometimes showed the willingness to respond to the questionnaires next to a phone or video call.

## 4. Discussion

Overall, the mouth-scanning procedures, delivery, and use of the PSPP were well accepted by both infants and parents. The feasibility of the workflow allowed plate adjustments and additional scans to produce newly adjusted PSPPs, thus extending the total duration time of their use. Parents’ response to the follow-up assessments was generally high, also indicating that the proposed workflow was viable for very young infants with DS.

The use of the SPP to improve mouth closure and tongue protrusion in children with DS, aiming to prevent secondary orofacial pathology associated with chronic tongue protrusion, is well documented in the literature [8,16,17,18,19,20,27]. The high acceptability of the digital workflow proposed in this study is thus a welcome contribution to support the use of the SPP, especially in infants with DS.

A comparison with the conventional impression-derived PSPP highlights that the proposed digital workflow offered the same features with new advantages. First, the introduction of the PSPP used in the current study provided a sense of security for parents/guardians, which allowed for an increased usage time, including during sleeping time. Additionally, the pacifier terminal favored lip sealing. Thus, the previously reported distinctive features of the impression-derived PSPP [21,22] were confirmed in our study. Notably, at no time after the PSPP delivery session did parents/guardians show any concerns during the use of the modified plate.

Second, in our study, there were some interruptions in the use of the PSPP related to periods of illness of the infant, the eruption of primary teeth, and temporary rejection of the PSPP by the child, along the lines of previous reports. It is known that children with DS are predisposed to a number of medical conditions that may manifest as frequent health problems and/or increased risk of recurrent episodes of illness and infections [31], and this was also observed in our study, leading to occasional discontinuation of PSPP use. Dental eruption has been reported to cause difficulties in SPP retention [18]. Eruption of the primary teeth in individuals with DS is usually delayed, with the first tooth often appearing between the ages of 12 and 14 months [1]. It was thus unsurprising that parents reported difficulties with the PSPP when teeth started to erupt, especially for the infants with extended use of the PSPP in our study. Moreover, as the nervous system matures and the child gets older, the tongue gains the ability to move more [32]. This improves the child’s capability to expel the plate, making its use more difficult and reducing the time of use with growing age, as also observed in our study.

Third, our study used a digital workflow that contrasts with the conventional impression-derived palatal plates. For the construction of the conventional palatal plate, it was necessary to make impressions of the edentulous dental arches using impression materials like alginate or elastomers. These materials are often associated with complications, such as increased risk of accidental ingestion, aspiration, and suffocation. Using the intraoral scanner to acquire models of the dental arches, as performed in the present study, eliminated these risks, offering a more suitable alternative without introducing any new risks. We only registered a small superficial lesion to the gingiva during mouth scanning in one of the participants. This is in line with results from studies using intraoral scanners in neonates and infants with craniofacial disorders [26]. Moreover, conventional impression techniques imply a single step, which can only be finished after the material achieves sufficient firmness before removal from the patient’s mouth. By contrast, intraoral scanning allows for the possibility to pause image acquisition whenever needed and resume at any point in time [33], a convenient feature used in the present study that allowed for scanning completion. However, it is recommended to use a scanner design that is well adjusted to the characteristics of the participants, to further improve their safety.

There is a lack of studies on the use of intraoral scanners and a digital workflow, especially in the care of infants and children with DS. Previous reports are still scarce and limited [26,34,35]. However, the use of intraoral scanners for constructing the SPP, as proposed in the current workflow, is also supported by studies showing that intraoral digital impressions are sufficiently accurate for fabricating scaffolds, retainers, and miniature connectors, and that they provide accurate measurements of the width of the alveolar arch and cleft defects [25,36,37]. Le Texier et al. [38] evaluated the accuracy of the IOS Dexis™ IS 3800 W and two other intraoral scanners for replicating a complete denture. For accuracy and precision evaluation using the ISO 5725-1 standard [39], specific to upper maxillary prostheses, the scanner showed an accuracy value of 106 ± 47 (mean ± standard deviation) and a precision value of 119 ± 24 (mean ± standard deviation). Nevertheless, differences between digital and plaster models of the soft palate and buccal vestibule areas were found in some of these studies, suggesting the need to further improve the accuracy of intraoral scanners.

Acquiring oral cavity models through digital scanners increases the options for processing, storing, and backing up the data obtained. It also allows for data control and transfer, facilitating the communication between all participants involved (dentist, child, parents/guardians, prosthetic laboratory). Furthermore, obtaining oral cavity data/models is less dependent on the child’s intrinsic characteristics, compared to the conventional methods. These advantages are highlighted in several studies conducted on children with cleft lip and palate [26,33,40,41], and different authors agree that the implementation of a digital workflow with these children offered greater efficiency to the entire process.

The duration of the scanning procedure is a further sensitive feature. In our study, the scanning duration of the maxilla was on average 482 s (8.2 min). In previous studies, different findings have been reported with a median scanning duration of the upper jaw of 129.5 s for children with DS, or a mean scanning time of 10.74 to 17.83 min in typically developing children, depending on the intraoral scanning procedure implemented [26,37]. Although differences in the scanners used and methodological procedures across studies might play a role, it is generally recognized that intraoral scanning of younger patients is a complex procedure characterized by difficulties in maneuvering the scanner head, leading to longer scanning duration. Moreover, oral scanning in children with DS presents additional difficulties because of the hypotonic muscles, protruded tongue, and hypersalivation [35,40]. Considering all these factors, the scanning duration results obtained in our study are suggested to reflect a further strength of the proposed workflow. Still, further improvements in intraoral scanning technology could be directed to provide a better response to the obstacles reported.

In addition to the results from the scanning and PSPP use, the viability and acceptability of the proposed workflow were also reflected in the responses to the follow-up sessions and language assessment questionnaires. Periodic video recordings of the clinical cases are a common procedure in palatal plate interventions [18,20,21]. We aimed for monthly video recordings, and the lowest response obtained was from one participant who completed only half of the video sessions. Given that many studies have used less stringent criteria, with recordings in intervals of three months (e.g., [20]), the rate of response obtained aligns with acceptable practices, allowing the subsequent assessment of oral-morphological variables through video registration. Unlike the evaluation of oral-morphological variables, which is standard practice in SPP interventions, the implementation of a procedure allowing for the assessment of language and communication abilities is a novel feature of the proposed workflow. Parental response rate to the language questionnaires was 100%. This highly successful procedure will allow for the evaluation of the impact of the PSPP on infants’ developing speech perception and speech production abilities. This is an important outcome, as it will allow for future studies to address whether oral-motor impairments constrain both the perception and production of speech along the lines of recent research on the auditory-motor link in infancy [15,42].

Despite the promising feasibility and acceptability of the proposed workflow, the current study has several limitations that call for further investigation. We examined the feasibility and acceptability of the new workflow in a small sample of infants with DS, referred by a single clinical institution, and with a varied frequency of use of the PSPP and limited follow-up duration (between 6 and 11 months). Future research should apply the workflow to larger samples, from different clinical contexts, including participants under diverse language and speech therapy interventions, with an extended follow-up duration, and using statistical metrics. Furthermore, future research should improve several features of the proposed workflow. First, advancing beyond qualitative assessments of PSPP fit, as in the current study, requires the development of non-invasive age-appropriate quantitative evaluation strategies. To our knowledge, there is no published study that provides quantitative in vivo measurements of adaptation (fit/intaglio-surface congruence) and retention force for palatal plates used in infants or young children. Existing quantitative methods are well-established for adult removable mucosa-supported prostheses (e.g., 3-D digital superimposition of scanned denture intaglio vs. reference casts, silicone-replica gap measurements, and mechanical dislodgement-force tests), but these have only been validated in adult edentulous jaws [43]. One study recently assessed the adaptation of feeding plates for infants with cleft palate using micro-computed tomography (micro-CT) to measure the 3-D volumetric space between the plate base and model, but the assessment was performed on plaster models—not intraorally [44]. Thus, although quantitative methods for adaptation and retention exist, they cannot be directly transferred to infant palatal plates given the anatomical, physiological, behavioral, and ethical constraints (small oral cavity, soft tissue compliance, patient cooperation, risk of gag reflex or aspiration), a challenge that future research needs to address. Second, although biocompatibility of the materials used in the manufacturing of the PSPP is guaranteed by manufacturers, analyses of the degree of polymerization and residual monomer levels are lacking, and further biocompatibility testing should be incorporated into future work. Third, our proposed workflow relied on the advantages and accuracy of intraoral scanners as reported in previous work [25,26,34,35,36,37,38], but a quantitative assessment of the accuracy and precision of the STL after processing by EXOCAD is still lacking. These limitations, together with the need for a cost-effectiveness analysis, constitute priorities for future research.

Last but not least, future work should compare the results across different intraoral scanner operators to enhance efficacy and best practices. Ways to improve intraoral scanner technology for younger patients and challenging clinical populations should also be explored. Importantly, as the new workflow is applied in interventions with more children, future research needs to determine its effects on oral morphology and auditory-motor language abilities in infants with DS, and progressively adjust the workflow to improve the effects obtained.

## 5. Conclusions

The current study described a new digital workflow for the construction and use of a modified palatal plate (PSPP) in infants with Down Syndrome and evaluated the feasibility and acceptability of the proposed workflow. The results underscored that the proposed digital workflow constitutes a viable and infant-friendly approach to the production and use of the PSPP device. The low risks and ease of implementation of the proposed workflow are suggested to facilitate the generalized use of therapeutic interventions with a PSPP, a step forward in the treatment and prevention of oral-motor dysmorphologies in infants.

## Figures and Tables

**Figure 1 dentistry-14-00026-f001:**
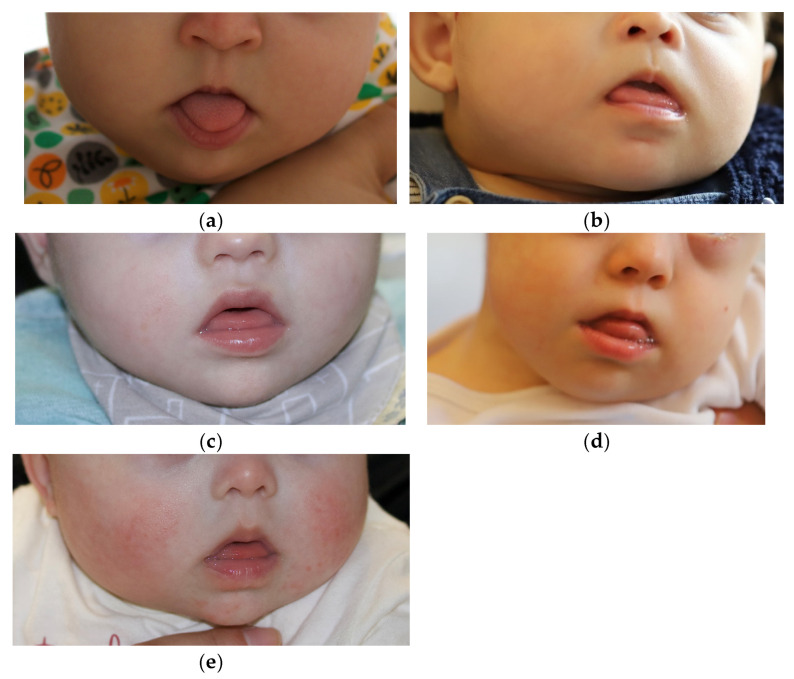
The five participants included in the study: (**a**) Participant 1; (**b**) Participant 2; (**c**) Participant 3; (**d**) Participant 4; (**e**) Participant 5.

**Figure 2 dentistry-14-00026-f002:**
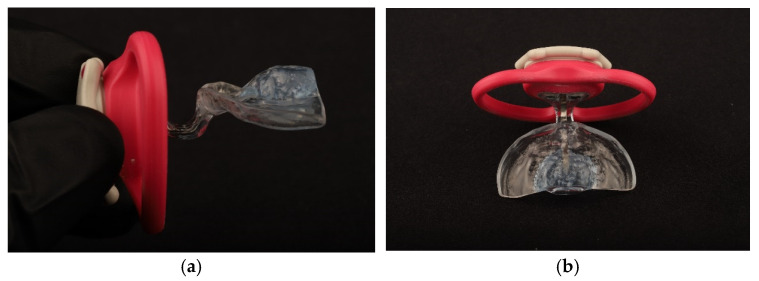
The pacifier stimulating palatal plate (PSPP): (**a**) lateral view; (**b**) upper view.

**Figure 3 dentistry-14-00026-f003:**
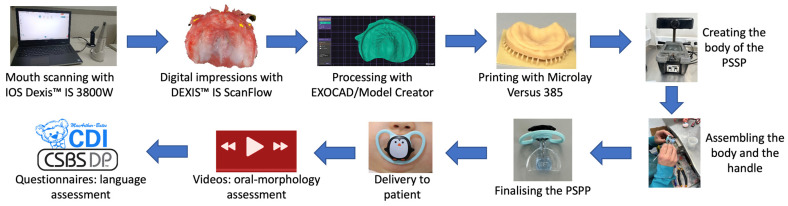
New digital workflow for the manufacturing and use of the modified SPP.

**Figure 4 dentistry-14-00026-f004:**
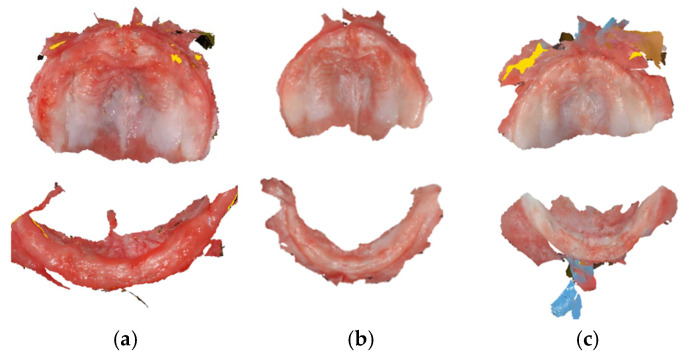
Oral scanning images displaying the upper and lower jaw across three participants: (**a**) Participant 1–5-month-old girl; (**b**) Participant 4–11-month-old boy; (**c**) Participant 5–6-month-old girl.

**Figure 5 dentistry-14-00026-f005:**
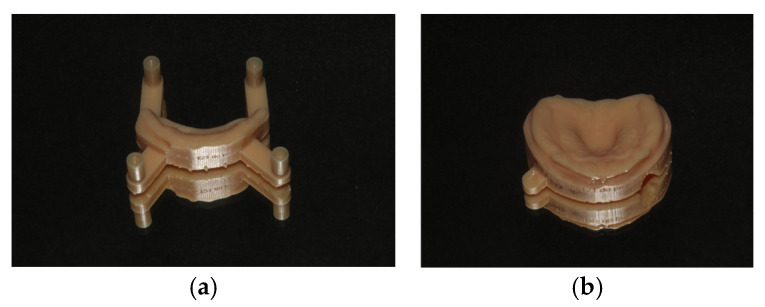
Digital model for Participant 1 (5-month-old girl): (**a**) digital model of the lower jaw; (**b**) digital model of the upper jaw.

**Figure 6 dentistry-14-00026-f006:**
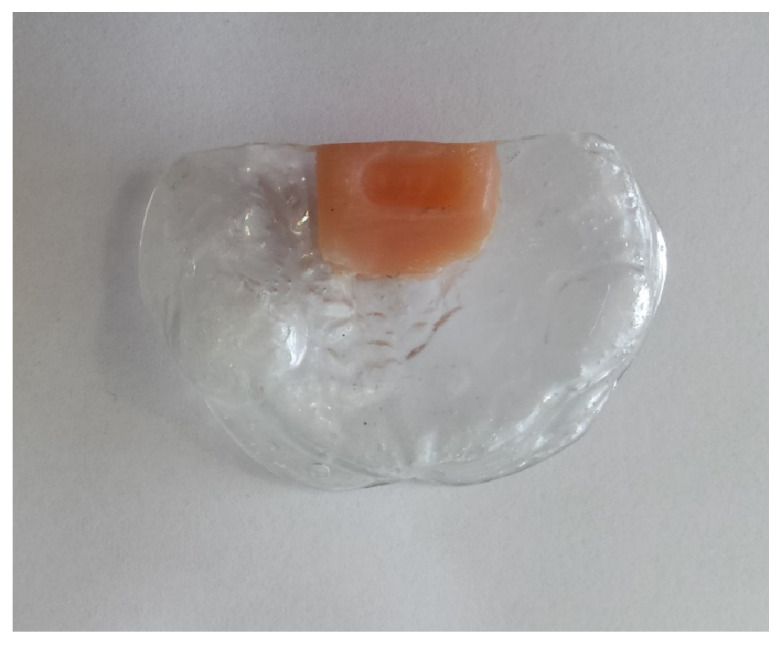
Body of the SPP, revealing the base and oval-shaped button.

**Figure 7 dentistry-14-00026-f007:**
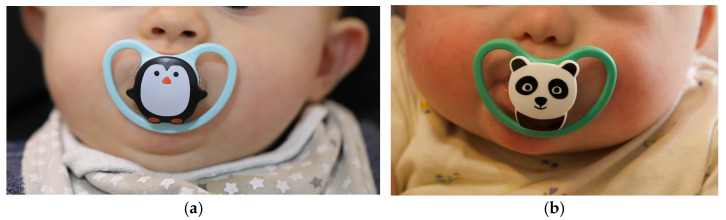
First day of use of the PSPP: (**a**) Participant 3 (7-month-old boy); (**b**) Participant 5 (6-month-old girl).

**Table 1 dentistry-14-00026-t001:** Variables for evaluating the viability and acceptability of the PSPP workflow.

Participant	Mouth Scanning Duration (Min)			Time Interval Scanning to Delivery (Days)	Total Duration of Use (Months)	Frequency of Use (Per Day)	Parental Rate Response to Follow-Up Contacts (Telephone and Video)	Parental Rate Response to Video Sessions	Parental Rate Response to CSBS	Parental Rate Response to CDI
	Upper Jaw	Lower Jaw	Total							
Participant 1	11	3	14	24	11	5 min to 1 h	91%	82%	100%	100%
Participant 2	12	6	18	14	6	15 min to 2 h	100%	67%	100%	100%
Participant 3	8	6	14	21	10	5 min to 30 min	100%	50%	100%	100%
Participant 4	7	5	12	19	10	1 h to 6 h	100%	70%	100%	100%
Participant 5	3	6	9	18	9	15 min to 30 min	78%	67%	100%	100%

## Data Availability

Given the restrictions imposed by the Ethics Committee, the detailed data from this study are available from the corresponding author, S.F., upon request.

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
