# Peer review of "New Digital Workflow for the Use of a Modified Stimulating Palatal Plate in Infants with Down Syndrome"

_dentistry, 2026, doi:10.3390/dj14010026_

Round 1

Reviewer 1 Report

Comments and Suggestions for Authors Missing or inconsistent use of definite/indefinite articles.
Examples:

1) “Trisomy 21, or Down Syndrome (DS), is most common genetic disorder among newborns.”
✅ “Trisomy 21, or Down Syndrome (DS), is the most common genetic disorder among newborns.”

2)“To manufacture SPP, it is necessary to take impressions of upper arch.”
✅ “To manufacture the SPP, it is necessary to take impressions of the upper arch.”

3) “The study proposes new digital workflow.”
✅ “The study proposes a new digital workflow.”

Slightly awkward or literal phrasing; better flow needed.
Examples:

1) “The PSPP was well adapted to the infant’s mouth in all cases.”
✅ “The PSPP fitted well in all infants.”

2) “Parents were very responsive to the monthly contacts, and the request to fill in the language questionnaires every 3 months.”
✅ “Parents responded actively to monthly contacts and to the three-month language questionnaires.”

3) “The study was developed within the P2LINK project…”
✅ “The study was conducted as part of the P2LINK project…”

Past and present tenses occasionally mixed.
Examples:

1)“The workflow is described and results were analyzed.”
✅ “The workflow was described and results were analyzed.”

2) “This modification addressed several issues and allows extended use of the SPP.”
✅ “This modification addressed several issues and allowed extended use of the SPP.”

Preposition use sometimes non-standard.
Examples:

1) “Increase of saliva production” → ✅ “Increase in saliva production.”

2)“Difference between digital and plaster models in soft palate area.” → ✅ “Difference between digital and plaster models of the soft palate area.”

3) “Promote tongue stimulation, tongue retraction, and stimulate oral-muscle function.”
✅ “Promote tongue stimulation, retraction, and overall oral-muscle function.”

Repetitive wording and unnecessary restatements.
Examples:

1) “The PSPP is formed of a body and a handle. For the body’s manufacturing, its base was created…”
✅ “The PSPP consists of two parts: the body and the handle. The base of the body was created…”

2) “All scans were performed in a single session and took, on average, 13.4 minutes to complete.”
✅ “Each scan was completed in a single session, taking an average of 13.4 minutes.”

Punctuation and Minor Typos

Common issues:
  • Missing commas after introductory phrases (“In the present study, …”)
  • Occasional double spaces after periods.
  • Overuse of parentheses — better replaced with commas in several sentences.
  • “months-old” → should be “month-old” (e.g., “11-month-old boy”).

  • Expand the comparison with other digital workflows in craniofacial and cleft palate rehabilitation, if available. This will help situate your work within the broader literature.
  • Limitations Section: Add a short paragraph emphasizing constraints such as small sample size, absence of quantitative metrics, and limited follow-up duration.
  • Figures and Tables: Ensure all figures are high resolution with consistent formatting and captions.
Comments on the Quality of English Language
  • A light professional proofreading focused on article usage, verb consistency, and punctuation.
  • No need for re-writing.A brief language edit would make the manuscript fully publication-ready.

Author Response

Comment 1: 1) “Trisomy 21, or Down Syndrome (DS), is most common genetic disorder among newborns.”
✅ “Trisomy 21, or Down Syndrome (DS), is the most common genetic disorder among newborns.”
2)“To manufacture SPP, it is necessary to take impressions of upper arch.”
✅ “To manufacture the SPP, it is necessary to take impressions of the upper arch.”
3) “The study proposes new digital workflow.”
✅ “The study proposes a new digital workflow.”
Slightly awkward or literal phrasing; better flow needed.
Examples:
1) “The PSPP was well adapted to the infant’s mouth in all cases.”
✅ “The PSPP fitted well in all infants.”
2) “Parents were very responsive to the monthly contacts, and the request to fill in the language questionnaires every 3 months.”
✅ “Parents responded actively to monthly contacts and to the three-month language questionnaires.”
3) “The study was developed within the P2LINK project…”
✅ “The study was conducted as part of the P2LINK project…”
Past and present tenses occasionally mixed.
Examples:
1)“The workflow is described and results were analyzed.”
✅ “The workflow was described and results were analyzed.”
2) “This modification addressed several issues and allows extended use of the SPP.”
✅ “This modification addressed several issues and allowed extended use of the SPP.”
Preposition use sometimes non-standard.
Examples:
1) “Increase of saliva production” → ✅ “Increase in saliva production.”
2)“Difference between digital and plaster models in soft palate area.” → ✅ “Difference between digital and plaster models of the soft palate area.”
3) “Promote tongue stimulation, tongue retraction, and stimulate oral-muscle function.”
✅ “Promote tongue stimulation, retraction, and overall oral-muscle function.”
Repetitive wording and unnecessary restatements.
Examples:
1) “The PSPP is formed of a body and a handle. For the body’s manufacturing, its base was created…”
✅ “The PSPP consists of two parts: the body and the handle. The base of the body was created…”
2) “All scans were performed in a single session and took, on average, 13.4 minutes to complete.”
✅ “Each scan was completed in a single session, taking an average of 13.4 minutes.”
Punctuation and Minor Typos
Common issues:

  • Missing commas after introductory phrases (“In the present study, …”)
  • Occasional double spaces after periods.
  • Overuse of parentheses — better replaced with commas in several sentences.

“months-old” → should be “month-old” (e.g., “11-month-old boy”)

Comments on the Quality of English Language

  • light professional proofreading focused on article usage, verb consistency, and punctuation. No need for re-writing.A brief language edit would make the manuscript fully publication-ready.

Response 1. Thank you very much for taking the time to review this manuscript. We have considered all your comments and suggestions, and performed a through proofreading and language editing. The amendments to our manuscript are identified by track changes and use of the red colour in the re-submitted file.

Comment 2: ‘Expand the comparison with other digital workflows in craniofacial and cleft palate rehabilitation, if available. This will help situate your work within the broader literature.’

Response 2: Thank you. We now added more studies that used intraoral scanners and a digital workflow in children in cleft lip and palate conditions, discussing their reported advantages.

Comment 3: ‘Limitations Section: Add a short paragraph emphasizing constraints such as small sample size, absence of quantitative metrics, and limited follow-up duration.’

Response 3: Thank you for this comment. We have added a few lines to the limitations section following your suggestions.

Comment 4: ‘Figures and Tables: Ensure all figures are high resolution with consistent formatting and captions.’

Response 4: This has been doubled checked.

Reviewer 2 Report

Comments and Suggestions for Authors

This paper reports on the development and clinical application of a modified Pacifier Stimulating Palatal Plate using a digital workflow. The goal is to improve oral motor function in infants and toddlers with Down syndrome.Since there is no comparative data with conventional methods, adding a mechanistic analysis would more clearly demonstrate this study's advantages.
The concerns raised after peer review are listed below.

1.Comparison with Conventional Impression-Taking Methods
The clinical effects of PSPP (tongue protrusion improvement, mouth closure, language development) cannot be clearly attributed to the digital workflow or the PSPP device itself. A comparison with conventional impression-derived PSPP, or mechanistic analysis showing how scanning accuracy improves clinical outcomes, would strengthen the evidence.

2.Visual Representation of Digital Workflow
An integrated flowchart showing the complete workflow is needed (DexisIS 3800W scanning → DEXIS IS ScanFlow and EXOCAD/Model Creator processing → Microlay Versus 385 printing → PSPP assembly → clinical assessment.)
This would improve clarity and reproducibility.

3.Quantitative Accuracy and Cost-Effectiveness Analysis
These data are essential for evaluating clinical feasibility and adoption.
Accuracy validation missing:
STL file precision after EXOCAD post-processing: no quantitative data
3D printer precision: layer thickness reported (385-405 nm) but not validated
Final product fit: only qualitative ("well-adapted"), no objective measurement
Cost-effectiveness analysis absent:
- Equipment costs, material costs, labor time not provided.
- No comparison with conventional methods.

Author Response

Comment 1: 'This paper reports on the development and clinical application of a modified Pacifier Stimulating Palatal Plate using a digital workflow. The goal is to improve oral motor function in infants and toddlers with Down syndrome.Since there is no comparative data with conventional methods, adding a mechanistic analysis would more clearly demonstrate this study's advantages. The concerns raised after peer review are listed below.'

Response 1: Thank you very much for taking the time to review this manuscript. We have carefully considered all the comments, and have made the necessary adjustments to our manuscript as identified by track changes and use of the red colour in the re-submitted file.

Comment 2: 'Comparison with Conventional Impression-Taking Methods

The clinical effects of PSPP (tongue protrusion improvement, mouth closure, language development) cannot be clearly attributed to the digital workflow or the PSPP device itself. A comparison with conventional impression-derived PSPP, or mechanistic analysis showing how scanning accuracy improves clinical outcomes, would strengthen the evidence.'

Response 2: We appreciate your comment. In the discussion, we compared the proposed digital workflow with the conventional impression-derived PSPP, highlighting that the previously reported distinctive advantages of the PSPP, namely the promotion of lip sealing and the increased sense of security in using the device, were also found in our study. In addition, the use of the IOS instead of the conventional impressions eliminated the risks associated with the impression materials without introducing new complications.

Comment 3: 'Visual Representation of Digital Workflow

An integrated flowchart showing the complete workflow is needed (DexisIS 3800W scanning → DEXIS IS ScanFlow and EXOCAD/Model Creator processing → Microlay Versus 385 printing → PSPP assembly → clinical assessment.) This would improve clarity and reproducibility.'

Response 3: We agree. We have now added a new figure (Figure 3) with a visual representation of the digital workflow.

Comment 4: 'Quantitative Accuracy and Cost-Effectiveness Analysis

These data are essential for evaluating clinical feasibility and adoption. Accuracy validation missing: STL file precision after EXOCAD post-processing: no quantitative data; 3D printer precision: layer thickness reported (385-405 nm) but not validated; Final product fit: only qualitative ("well-adapted"), no objective measurement; Cost-effectiveness analysis absent:- Equipment costs, material costs, labor time not provided; - No comparison with conventional methods.'

Response 4: Thank you for these comments. We provided as much information as possible, including 3D precision, final product fit, and comparison with conventional methods.

Reviewer 3 Report

Comments and Suggestions for Authors

The article titled “New digital workflow for the use of a modified Stimulating Palatal Plate in infants with Down Syndrome” is well presented by the authors, however I have few suggestions.

Abstract:

  1. This section does not give the gist of entire content of the paper
  2. The authors must include more details in the patient section.
  3. There is no mention of the study design in the title as well as abstract

Introduction:

  1. Introduction is well written; however, the authors need to add more details on the reported advantages of this devices compared to the traditional. They have mention that pervious study did not find any additional benefits.
  2. Add note on the patient compliance to these devices. As they are monitored by parents or guardians
  3. Authors need to mention the rationale for this paper, the need for this model.

Methods:

  1. Kindly mention more details on the sample, is this sufficient to generalize?
  2. Why was the study limited to sample from one single institution? Kindly justify as it will affect the generalizability of the results

Discussion:

  1. Authors should consider more articles for discussion
  2. Is there lack of similar studies in the literature
  3. Mention the limitations of the study
  4. Add note on patient compliance.

Conclusions:

  1. Conclusions must be based on the objective of this paper.

Author Response

Comment 1: 'The article titled “New digital workflow for the use of a modified Stimulating Palatal Plate in infants with Down Syndrome” is well presented by the authors, however I have few suggestions'

Response 1: Thank you for your assessment of our manuscript, and your useful suggestions. We have carefully considered all the comments and made the necessary adjustments to our manuscript as identified by track changes and use of the red colour in the re-submitted file.

Comment 2: 'Abstract:

  1. This section does not give the gist of entire content of the paper
  2. The authors must include more details in the patient section.
  3. There is no mention of the study design in the title as well as abstract'

Response 2: We have revised the abstract in line with your suggestions and also per the suggestions of another reviewer, within the strict word limits imposed by the journal.

Comment 3: 'Introduction:

  1. Introduction is well written; however, the authors need to add more details on the reported advantages of this devices compared to the traditional. They have mention that pervious study did not find any additional benefits.
  2. Add note on the patient compliance to these devices. As they are monitored by parents or guardians
  3. Authors need to mention the rationale for this paper, the need for this model.'

Response 3: Thank you for raising these points. We have now clarified that the modified SPP was found in a previous study to offer additional benefits when compared to the traditional plate. One of the benefits related to patient compliance. We have also made clear what the rationale for the proposal of the new workflow is, and why it is needed.

Comment 4: 'Methods: 

  1. Kindly mention more details on the sample, is this sufficient to generalize?
  2. Why was the study limited to sample from one single institution? Kindly justify as it will affect the generalizability of the results'

Response 4: Thank you for your comment. We have now included more information on the sample and on the rationale behind the sample used in the study. We have also included the limitation of using a sample from a single institution in the factors discussed in the paragraph on the study limitations.

Comment 5: 'Discussion:

  1. Authors should consider more articles for discussion
  2. Is there lack of similar studies in the literature
  3. Mention the limitations of the study
  4. Add note on patient compliance.'

Response 5: We appreciated your comments and changed the discussion section accordingly. We recognized the lack of similar studies in the literature, and added more studies that used intraoral scanners and a digital workflow in children, in cases like the cleft lip and palate conditions, discussing their reported advantages. Moreover, we included an extended paragraph on the limitations of our study, also per advice of another reviewer.

Comment 6: 'Conclusions: 

  1. Conclusions must be based on the objective of this paper.'

Response 6: Thank you. The conclusions were rewritten to directly reflect the objective of the paper.

Reviewer 4 Report

Comments and Suggestions for Authors

The paper is well structured and the procedures are described in detail. Here are some suggestions for the authors for the best presentation of the paper.

In Summary/methods should be mentioned how the developing language and communication abilities and the acceptability of the device were assessed (video captures and parental questionnaires).

Introduction . An additional commend should be made for the design of older devices’  that have been proposed for infants with Down Syndrome and practices in conjunction,  like help of speech professionals and physiotherapists.

Methods: Had the participating infants additionally language intervention, and oral motor and sensory stimulation provided by speech therapists or physiotherapists? If yes, was there some impact on the results of the study?

The use of a control group with an identical plate constructed without the use of the digital workflow that is proposed would be the best method for assessing the aim of the study.

In Discussion the limitations of the study should be clearly discussed in more detail, specifically the frequency of use per day for each participant and possible help on speech features by speech professionals

Author Response

Comment 1: 'The paper is well structured and the procedures are described in detail. Here are some suggestions for the authors for the best presentation of the paper.'

Response 1: Thank you for your positive assessment of our manuscript, and the useful suggestions. We have carefully considered all the comments, and have made the necessary adjustments to our manuscript as identified by track changes and use of the red colour in the re-submitted file.

Comment 2: 'In Summary/methods should be mentioned how the developing language and communication abilities and the acceptability of the device were assessed (video captures and parental questionnaires).'

Response 2: Thank you for pointing this out. This has been done.

Comment 3: 'Introduction . An additional commend should be made for the design of older devices’  that have been proposed for infants with Down Syndrome and practices in conjunction,  like help of speech professionals and physiotherapists.'

Response 3: We have included this additional information.

Comment 4: 'Methods: Had the participating infants additionally language intervention, and oral motor and sensory stimulation provided by speech therapists or physiotherapists? If yes, was there some impact on the results of the study?'

Response 4: Thank you for raising this point. We have now included more information on the participating infants and on the rationale behind the characteristics of the sample used in the study. Our aim was to test the feasibility and acceptability of the new digital workflow in a diverse sample of participants (age, gender, participation or not in language intervention or speech therapy). Importantly, this diversity had no impact on the feasibility and acceptability results.

Comment 5: 'The use of a control group with an identical plate constructed without the use of the digital workflow that is proposed would be the best method for assessing the aim of the study.'

Response 5: Thank you for this point. A modified palatal plate constructed without the use of a digital workflow was already tested in a previous study, as discussed in the Introduction section. The results then obtained, in comparison with the conventional plate, led us to use the modified plate in the current study, to which we added the digital workflow. The potential advantages of the digital workflow have been outlined in the Introduction and now discussed in detail in the revised Discussion section.

Comment 6: 'In Discussion the limitations of the study should be clearly discussed in more detail, specifically the frequency of use per day for each participant and possible help on speech features by speech professionals'

Response 6: We appreciated you comment. We have included an extended paragraph on the limitations of our study, also per advice of another reviewers.

Round 2

Reviewer 2 Report

Comments and Suggestions for Authors

I reviewed the first draft and suggested three topics: (1) comparison with conventional methods and mechanistic analysis; (2) workflow flowchart; (3) quantitative accuracy and cost-effectiveness data.
In the revised manuscript, only Comment 2 (workflow flowchart) has been adequately addressed. Comment 1 remains partially addressed, and Comment 3 remains unaddressed. The Limitations section does not adequately explain the omission of precision verification and cost analysis. The authors must explicitly state why these data were not collected, or designate them as future research priorities.
Critical Issues
1. Weak Theoretical Basis for Oral-Auditory-Motor Link
The proposed mechanism (oral-motor dysfunction affects speech perception) lacks empirical verification specific to Down syndrome.
Confounding factor not controlled: Participants 3 and 4 received concurrent speech therapy during PSPP use. Language improvements cannot be attributed solely to PSPP.
Methods 2.3.4 explicitly limits analysis to "feasibility and acceptability," not efficacy.
The authors must: Reframe language outcome claims as exploratory, or explicitly address and control for concurrent therapies.
2. Missing Material Safety and Precision Verification
STL precision after EXOCAD processing: Not quantified.
Final PSPP fit: Only qualitative ("well-adapted"); no objective measurement.
Material biocompatibility: No ISO compliance testing. Degree of polymerization and residual monomer levels unmeasured.
Since the workflow is used in infants, formal material safety validation is essential before clinical implementation.
The authors must provide quantitative accuracy and ISO biocompatibility data, or explicitly acknowledge any limitations that require resolution.

Author Response

Comments 1: I reviewed the first draft and suggested three topics: (1) comparison with conventional methods and mechanistic analysis; (2) workflow flowchart; (3) quantitative accuracy and cost-effectiveness data.
In the revised manuscript, only Comment 2 (workflow flowchart) has been adequately addressed. Comment 1 remains partially addressed, and Comment 3 remains unaddressed. The Limitations section does not adequately explain the omission of precision verification and cost analysis. The authors must explicitly state why these data were not collected, or designate them as future research priorities.

Response 1: Thank you very much for your careful consideration of the revised version of our manuscript. We have done our best to address all your comments, and have made the necessary adjustments to our manuscript as identified by the use of the blue colour in the re-submitted file.

Comment 2: 1. Weak Theoretical Basis for Oral-Auditory-Motor Link. The proposed mechanism (oral-motor dysfunction affects speech perception) lacks empirical verification specific to Down syndrome.

Response 2: Thank you for this comment. We have rewritten the part on the oral-auditory-motor link in Down Syndrome to reflect that this is still on open question that merits empirical verification (kindly see p.2 and p.9).

Comment 3: Confounding factor not controlled: Participants 3 and 4 received concurrent speech therapy during PSPP use. Language improvements cannot be attributed solely to PSPP.

Response 3: Thank you for raising this point. We are aware that this is a key factor to be taken into account in any future analysis of language outcomes, and apologize for not having made it clear from the outset. As now explicitly mentioned, this factor would be taken into account in future analyses of infants’ developing language abilities (which are beyond the scope of the present study).  Kindly see, for example, p.3-4 (lines 141-142), and p.9, lines 339-354.

Comment 4: Methods 2.3.4 explicitly limits analysis to "feasibility and acceptability," not efficacy.

Response 4: Thank you. Indeed, as we clearly stated in the introduction, in the methods, and in the discussion and conclusion, our central goal is to evaluate the feasibility and acceptability of the proposed workflow.

Comment 5: The authors must: Reframe language outcome claims as exploratory, or explicitly address and control for concurrent therapies.

Response 5: Thank you. This has been done. Kindly see our answers to the two previous comments, namely comment 3.

Comment 6: Missing Material Safety and Precision Verification
STL precision after EXOCAD processing: Not quantified.

Response 6: We appreciate this comment. We have now added a reference to a further study on accuracy and precision evaluation of intraoral scanners, and in particular the scanner used in the present study (kindly see the discussion section, p. 12). Additionally, in the limitations section, we acknowledged that our proposed workflow lacks a quantitative assessment of the accuracy and precision of the STL after processing by EXOCAD, as well as a cost-effectiveness analysis, which are among the priorities for future research (discussion, p.14).

Comment 7: Final PSPP fit: Only qualitative ("well-adapted"); no objective measurement.

Response 7: We thank the reviewer for following up on this point. We have now further clarified all the qualitative assessments made, in line with previous work on stimulating palatal plates in infants (please kindly see section 2.3.2, p.8, and section 3.2., p.11). We have explicitly stated that only a qualitative assessment was used. In the limitations section, we have highlighted that using a qualitative assessment approach is a limitation of our study, and that in future work non-invasive quantitative measures adapted to this population should be sought and used (discussion, pp.13-14).

Comment 8: Material biocompatibility: No ISO compliance testing. Degree of polymerization and residual monomer levels unmeasured. Since the workflow is used in infants, formal material safety validation is essential before clinical implementation. The authors must provide quantitative accuracy and ISO biocompatibility data, or explicitly acknowledge any limitations that require resolution.

Response 8: Thank you for raising these issues. We have added all the information available to us on material biocompatibility (kindly see section 2.3.2, p.7). We have also explicitly acknowledged that analyses of degree of polymerization and residual monomer levels, and further biocompatibility testing should be incorporated into future work (discussion, p.14).

Round 3

Reviewer 2 Report

Comments and Suggestions for Authors

Taking impressions from children with Down's syndrome can be challenging due to risks such as aspiration, but demonstrating the safety and acceptability of this process using IOS has significant clinical value. Rather than refuting the reviewers' concerns about 'quantifying accuracy' and 'effects on long-term language development', the authors candidly acknowledge these as 'limitations of this study'. This allows readers to interpret the results realistically. A notable aspect of this manuscript is the shift in the study's focus from 'proving efficacy' to 'demonstrating the feasibility of a new workflow'. Acceptance is therefore recommended.